# NK Cells Lose Their Cytotoxicity Function against Cancer Stem Cell-Rich Radiotherapy-Resistant Breast Cancer Cell Populations

**DOI:** 10.3390/ijms22179639

**Published:** 2021-09-06

**Authors:** Hana Jin, Hye Jung Kim

**Affiliations:** Department of Pharmacology, Institute of Health Sciences, College of Medicine, Gyeongsang National University, 816 Beongil 15 Jinjudaero, Jinju 52727, Korea; hanajin.kr@daum.net

**Keywords:** cancer stem cells, radiotherapy-resistance, triple negative breast cancer, NK cells, immune escape

## Abstract

Cancer stem cells (CSCs) can be induced from differentiated cancer cells in the tumor microenvironment or in response to treatments and exhibit chemo- and radioresistance, leading to tumor recurrence and metastasis. We previously reported that triple negative breast cancer (TNBC) cells with acquired radioresistance exhibited more aggressive features due to an increased CSC population. Therefore, here, we isolated CSCs from radiotherapy-resistant (RT-R)-TNBC cells and investigated the effects of these CSCs on tumor progression and NK cell-mediated cytotoxicity. Compared to MDA-MB-231 and RT-R-MDA-MB-231 cells, CD24^−/low^/CD44^+^ cells isolated from RT-R-MDA-MB-231 cells showed increased proliferation, migration and invasion abilities, and induced expression of tumor progression-related molecules. Moreover, similar to MDA-MB-231 cells, CD24^−/low^/CD44^+^ cells recruited NK cells but suppressed NK cell cytotoxicity by regulating ligands for NK cell activation. In an in vivo model, CD24^−/low^/CD44^+^ cell-injected mice showed enhanced tumor progression and lung metastasis via upregulation of tumor progression-related molecules and altered host immune responses. Specifically, NK cells were recruited into the peritumoral area tumor but lost their cytotoxicity due to the altered expression of activating and inhibitory ligands on tumors. These results suggest that CSCs may cause tumor evasion of immune cells, resulting in tumor progression.

## 1. Introduction

Tumors are composed of a highly heterogeneous cell population that includes cancer stem cell-like cells (CSCs). Many researchers have suggested the important roles of CSCs in tumor progression in various types of cancers [1,2,3,4,5]. CSCs represent a small population of cancer cells that share features in common with normal stem cells and appear to mimic the developmental program of corresponding normal tissue stem cells, although in an incomplete and disorganized manner. CSCs are also called ‘tumor-initiating cells’, distinct from nonmalignant normal stem cells, and show a high self-renewal capacity, the propensity to differentiate into actively proliferating tumor cells, the ability to repair DNA damage, and low levels of reactive oxygen species (ROS). In addition, the small population of CSCs may determine the biological properties of cancers, including the therapeutic response. Emerging evidence indicates that the CSC population is often highly resistant to chemotherapy or radiotherapy (RT). CSCs can survive after conventional anticancer therapy, and the survival of even one CSC after treatment can lead to tumor recurrence, which has emerged as the major risk factor for death in patients with cancer [6,7,8,9].

RT is one of the major treatment methods for cancer, and approximately one-half to two-thirds of all cancer patients receive some form of RT as a part of their primary treatment regimen [10]. RT increases the intracellular ROS level in cancer cells, causing DNA damage. Therefore, cell cycle progression is delayed in these cancer cells to facilitate DNA repair via initiation of the DNA damage response [11,12,13]. When RT-induced DNA damage becomes irreparable, the DNA damage response induces apoptosis, senescence or mitotic catastrophe in cancer cells [14]. Therefore, a successful RT with a sufficient dose of irradiation not only suppresses further cancer cell growth but also kills cancer cells. However, the resistance of cancer cells to RT and the residual cancer mass in the body are key causes of recurrence as well as metastasis, adversely affecting the prognosis of patients with cancer. Therefore, in our previous study, we generated RT-resistant (RT-R) breast cancer (BC) cell lines and determined that RT-R-BC cells, especially RT-R-BC cells derived from triple-negative breast cancer (TNBC), exhibited enhanced resistance to chemotherapy and promoted invasive properties. TNBC is difficult to treat due to its aggressive features and absence of hormone receptors. In addition, we further found that the aggressive properties were enhanced due to the increased CSC population among RT-R-TNBC cells compared to the parental cells or other RT-R-non TNBC cells [15], suggesting that an increased CSC population among cancer cells might be related to the acquisition of resistance to anticancer therapies.

In general, the normally functioning immune system, which is a complex network of cells, tissues, and organs that can be divided into the innate and adaptive immune systems, contributes to the recognition and removal of foreign pathogens as well as tumors [16]. However, interestingly, the immune cells in the tumor microenvironment (TME) show a functional change and are related to the growth and metastasis rather than the removal of tumor cells. In fact, a clinical study reported that compared to a low neutrophil-to-lymphocyte ratio (NLR) (<4), a high NLR (≥4) at diagnosis was associated with poor performance status, advanced stage, and a lower response rate [17]. Moreover, we reported that tumor-associated macrophages (TAMs) are recruited into and accumulate in the TME and are involved in cancer cell growth and metastasis through interactions with cancer cells [18]. Therefore, the function of immune cells in the TME is expected to be changed from removing cancer cells to playing a role in facilitating their growth and metastasis.

In particular, natural killer (NK) cells, a subset of innate lymphoid cells, are the most efficient effector cells among cytolytic lymphocytes, exhibiting natural cytotoxicity against primary tumor cells and suppressing metastasis through inhibition of cancer cell proliferation, migration, and colonization to distant organs [19]. However, the NK cell function is generally inhibited in the TME and provides conditions favorable for tumor progression. The TME produces and accumulates soluble regulators that negatively regulate the proliferation, maturation, and effector function of NK cells [20]. In addition, these immunosuppressive molecules act on NK cells not only directly but also indirectly via stimulation of other immune cells, such as antigen-presenting cells (APCs), regulatory T cells (Tregs), and myeloid-derived suppressor cells (MDSCs), to produce additional immunosuppressive factors [21]. Interestingly, in the TME, tumor cells per se induce immunosuppression through selection for cells that develop mechanisms of long-term immune escape, leading to a lack of capacity for immune-mediated tumor clearance, a process termed cancer immunoediting. This process is defined by phases of elimination, equilibrium, and escape: Immunosurveillance occurs in the first phase, while selection of tumor variants occurs in the second phase, eventually resulting in immune escape, which has been considered essential for tumorigenesis and metastasis [22,23,24].

As mentioned above, since compared to the parental cells, RT-R-TNBC cells exhibit more aggressive properties in terms of tumor growth, invasion, and metastasis and harbor an increased CSC population, we sought to determine whether the increased CSC population among RT-R-TNBC cells plays a role in the evasion of RT-R-TNBC from the NK cell-mediated immune response. Therefore, in this study, we isolated CSCs from RT-R-MDA-MB-231 cells and investigated the properties of these CSCs and whether they regulate NK cell-mediated cytotoxicity and therapeutic resistance.

## 2. Results

### 2.1. Breast Cancer Cells with Acquired Radioresistance Contain a Larger Population of Breast Cancer Stem Cells (BCSCs) and Increased Proliferation, Migration, and Invasion Abilities

First, we compared the populations of CD24^−/low^/CD44^+^ BCSCs between two cells MDA-MB-231 and RT-R-MDA-MB-231 and CD24^−/low^/CD44^+^ cells isolated from RT-R-MDA-MB-231 cells. The population of CD24^−/low^/CD44^+^ cells among RT-R-MDA-MB-231 cells was approximately two-fold greater than that among MDA-MB-231 cells (Figure 1A). Then, we examined the properties of CD24^−/low^/CD44^+^ cells related to tumor progression by comparing MDA-MB-231 cells and RT-R-MDA-MB-231 cells harboring different proportions of CD24^−/low^/CD44^+^ cells with the CD24^−/low^/CD44^+^ cells isolated from RT-R-MDA-MB-231 cells. As shown in Figure 1B–E, compared to MDA-MB-231 cells, RT-R-MDA-MB-231 cells showed increased cell proliferation, colony formation, migration and invasion abilities, but CD24^−/low^/CD44^+^ cells isolated from RT-R-MDA-MB-231 cells showed the greatest enhancement of abilities in these assays.

### 2.2. The Increased Population of CD24^−/low^/CD44^+^ Cells among RT-R-BCs Induces the Expression Levels of Tumor Progression-Related Proteins

Epithelial–mesenchymal transition (EMT) is the process by which epithelial cells transdifferentiate into motile mesenchymal cells and contributes to cancer cell migration/invasion and tumor progression [25]. Therefore, we examined the expression levels of EMT-related molecules in CD24^−/low^/CD44^+^ cells compared to MDA-MB-231 cells and RT-R-MDA-MB-231 cells. Compared to MDA-MB-231 cells, RT-R-MDA-MB-231 cells showed decreased levels of E-cadherin, an epithelial marker, and increased levels of the mesenchymal markers N-cadherin, β-catenin, and Snail. Interestingly, CD24^−/low^/CD44^+^ cells exhibited lower expression levels of E-cadherin and higher expression levels of N-cadherin, β-catenin, and Snail than either MDA-MB-231 or RT-R-MDA-MB-231 cells (Figure 2A; Appendix A). In addition, we compared the expression levels of ESM-1, HIF-1α, and LOX in these three cell populations, based on a report indicating that the HIF-1α/LOX pathway plays a key role in promoting breast tumor growth and metastasis [26] and our previous observation that overexpression of ESM-1 in RT-R-breast cancer cells was involved in enhanced tumor progression [27]. ESM-1 expression was increased in RT-R-MDA-MB-231 cells compared to MDA-MB-231 cells and was even induced stronger in CD24^−/low^/CD44^+^ cells (Figure 2B; Appendix A). Moreover, both HIF-1α expression and LOX secretion were increased in RT-R-MDA-MB-231 cells compared to MDA-MB-232 cells and were increased even further in CD24^−/low^/CD44^+^ cells isolated from RT-R-MDA-MB-231 cells (Figure 2C,D; Appendix A).

### 2.3. CD24^−/low^/CD44^+^ Cells Recruit NK Cells but Suppress Their Cytotoxicity through Modulation of the Levels of NK Cell Function-Related Ligands

Reports indicate that NK cells are a major type of infiltrating immune cell in breast cancer and prevent tumorigenesis and subsequent metastasis [28,29]. However, BCSCs are reported to be resistant to NK cell cytotoxicity and contribute to lung metastasis by regulating the expression of ligands for NK cell receptors [30]. Therefore, we sought to determine whether NK cells are recruited to BCSCs, whether their function is modified, and if so, how this change of function is mediated. First, we examined the migration of NK cells toward MDA-MB-231 cells and CD24^−/low^/CD44^+^ cells. In Figure 3A, NK-92MI cell migration across the Transwell membrane toward the conditioned medium from MDA-MB-231 and CD24^−/low^/CD44^+^ cells were induced stronger compared to that toward the fresh medium, but no significant difference was observed between the migration toward the conditioned medium from MDA-MB-231 and CD24^−/low^/CD44^+^ cells. Then, we further investigated the cytotoxicity of NK cells against MDA-MB-231 and CD24^−/low^/CD44^+^ cells. Interestingly, NK-92MI cells exhibited an increased cytotoxic effect on target MDA-MB-231 cells as the ratio of NK-92MI cells to target cells increased. However, NK-92MI cells showed a decreased cytotoxic effect on CD24^−/low^/CD44^+^ cells compared to MDA-MB-231 cells at all ratios (Figure 3B). Perforin and granzyme B produced by NK cells play a key role in NK cell-mediated cytotoxicity [31]. Therefore, we measured the levels of secreted perforin and granzyme B in the medium from NK-92MI cells cocultured with MDA-MB-231 cells or CD24^−/low^/CD44^+^ cells and found that the levels of secreted perforin and granzyme B were lower in the medium from NK-92MI cells cocultured with CD24^−/low^/CD44^+^ cells than in the medium from NK-92MI cells cultured with MDA-MB-231 cells (Figure 3C,D).

To investigate the cause of the altered cytotoxicity of NK cells, we examined the expression of ligands for NK cell receptors, which regulate the functional activity of NK cells, on breast cancer cells. HLA class I, E (HLA-E) molecules are nonclassical human MHC class I molecules that promote tolerance to NK cytotoxicity, and MHC class I chain-related molecules A/B (MICA/B) enhance the functions of NK and T cells [32]. As seen in Figure 4A,B, the expression levels of MICA/B, the ligands for the activating NKG2D receptor, were decreased but the expression of HLA-E, the ligand for the inhibitory NKG2A receptor, was increased in RT-R-MDA-MB-231 and CD24^−/low^/CD44^+^ cells compared to MDA-MB-231 cells (Figure 4A,B; Appendix A). Since a reduction in the expression of NKG2D ligands on the cancer cell surface, which is mediated by the cleavage activity of a disintegrin and metalloproteinase domain containing protein-10 (ADAM-10), constitutes an immune escape mechanism that impairs NK cell recognition [33], we next determined the ADAM-10 level in MDA-MB-231, RT-R-MDA-B-231, and CD24^−/low^/CD44^+^ cells. Interestingly, the level of the active form of ADAM-10 was increased in RT-R-MDA-B-231 and CD24^−/low^/CD44^+^ cells compared to MDA-MB-231 cells (Figure 4C; Appendix A). However, we did not observe significant differences in the levels of MICA/B, HLA-E, and ADAM-10 between RT-R-MDA-MB-231 and CD24^−/low^/CD44^+^ cells. Therefore, these results suggest that breast cancer cells containing a larger BCSC population suppress the cytotoxicity of NK cells by impairing NK cell recognition of cancer cells and inducing inhibitory signaling in NK cells.

### 2.4. CD24^−/low^/CD44^+^ Cells Exhibit Enhanced Aggressive Features Supporting the Progression and Metastasis of Breast Cancer In Vivo

Finally, we confirmed the effect of MDA-MB-231, RT-R-MDA-MB-231, and CD24^−/low^/CD44^+^ cells on tumor progression in an in vivo xenograft model in mice. Six-week-old female BALB/c nude mice were injected subcutaneously with MDA-MB-231, RT-R-MDA-MB-231, and CD24^−/low^/CD44^+^ cells, and tumor volumes and body weights were measured three times weekly beginning on the 7th day after injection, as described in the Materials and Methods section. As shown in previous studies [18,34], the tumor volumes were not significantly increased in MDA-MB-231 cell-injected mice, since MDA-MB-231 human breast cancer cells cannot overcome the immune barrier in mice. However, the tumor volumes were significantly increased in RT-R-MDA-MB-231 cell-injected mice compared to MDA-MB-231 cell-injected mice, and CD24^−/low^/CD44^+^ cell-injected mice showed even greater enhancement of tumor growth than RT-R-MDA-MB-231 cell-injected mice (Figure 5A,B). The body weights were similar among the three groups (Figure 5C). However, metastasis to the lung (Figure 5D) was induced in the RT-R-MDA-MB-231 cell-injected group relative to the MDA-MB-231 cell-injected group and was greatly increased in the CD24^−/low^/CD44^+^ cell-injected group.

In addition, we examined the expression levels of EMT- and tumor progression-related molecules and determined that E-cadherin expression was higher in MDA-MB-231 tumor tissue than in tissues from the other groups. However, N-cadherin, β-catenin, Snail, and ESM-1 expression levels were increased in RT-R-MDA-MB-231 tumor tissue compared to MDA-MB-231 tumor tissue and were even higher in CD24^−/low^/CD44^+^ tumor tissue (Figure 6A,B). Moreover, the levels of tumor metastasis-related molecules, HIF-1α expression, and LOX secretion were increased in RT-R-MDA-MB-231 tumor tissue compared to MDA-MB-231 tumor tissue and were even higher in CD24^−/low^/CD44^+^ tumor tissue (Figure 6C,D). Furthermore, we investigated NK cell infiltration and the expression of NK cell activating or inhibitory ligands in tumor sections from the animal model and found that NK cell infiltration into tumor tissues was similar among the groups (Figure 7A). The NK cell activating ligand MICA/B showed higher expression in the MDA-MB-231 tumor tissue than in RT-R-MDA-MB-231 or CD24^−/low^/CD44^+^ tumor tissue (Figure 7B). However, the expression of the NK cell inhibitory ligand HLA-E exhibited the opposite pattern, with lower expression in MDA-MB-231 tumor tissue than in RT-R-MDA-MB-231 or CD24^−/low^/CD44^+^ tumor tissue (Figure 7C). In addition, the ADAM-10 expression level was increased in RT-R-MDA-MB-231 tumor tissue compared to MDA-MB-231 tumor tissue and was even higher in CD24^−/low^/CD44^+^ tumor tissue (Figure 7D). Interestingly, Figure 7E shows that MDSCs, Gr-1-positive cells were highly increased in RT-R-MDA-MB-231 tumor tissue compared to MDA-MB-231 tumor tissue and were even higher in CD24^−/low^/CD44^+^ tumor tissue (Figure 7E). These results suggest that BCSCs present in high proportions in the RT-R-MDA-MB-231 cell population exhibit aggressive features promoting breast tumor growth and metastasis by increasing the expression of tumor progression-related molecules and altering host immune responses.

## 3. Discussion

Tumor heterogeneity, the inclusion of various cell populations within a single tumor, makes research and treatment for cancers, including breast cancer, difficult and complex [35]. Two models have been proposed to explain tumor heterogeneity: (i) All cells in tumors undergo clonal evolution to acquire genetic alterations and develop a malignant phenotype. Alternatively, (ii) a subset of cells in tumors act as multifunctional progenitors to promote tumor growth, suggesting the concept of CSCs [2,36]. Specifically, CSCs are virtually resistant to RT and cytotoxic chemotherapy and may contribute to treatment resistance and tumor relapse [37]. RT has been a main therapeutic option for cancer due to its beneficial effects of suppressing cancer cell progression and killing cancer cells by inducing apoptosis and mitotic catastrophe [10,11,12,13,14]. However, several types of cancer recur after RT, and the RT response differs considerably among patients. Interestingly, recent views related to tumor progression suggest that a small population of CSCs may determine the biological properties of cancers, including the therapeutic response [38]. Although the mechanisms underlying the radioresistance of CSCs remain unclear, several studies have suggested that increased self-renewal activity, enhanced DNA repair ability, and decreased DNA damage induction by a reduction in ROS levels may be associated with increased radioresistance of CSCs [6,39]. In particular, the pathways that regulate the self-renewal capacity of normal stem cells include the WNT/β-catenin, Notch, and Snail/Slug pathways, which have also been implicated in the radioresistance of CSCs [40,41,42]. In our previous study, we determined that TNBC cells with acquired RT resistance exhibited significantly increased expression levels of Notch-4, β-catenin, and Snail, which are involved in self-renewal as well as tumor progression, compared to those in the parental cells [15]. In addition, interestingly, we found a higher population of CD24^−/low^/CD44^+^ cells among RT-R-TNBC cells than among TNBC cells, and RT-R-TNBC cells showed more aggressive tumorigenic properties related to tumor cell growth, migration and invasion than the parental breast cancer cells [15]. CSCs are identified based on their functional activity (self-renewal and serial tumor progression) and phenotypic markers (CD44^+^/CD24^−^ signature and aldehyde dehydrogenase-1 (ALDH) activity) [43]. CD44 has been positively associated with stem cell-like characteristics, and CD24 is related to differentiated epithelial features [44]. Based on these observations, we hypothesized that the acquisition of RT resistance by TNBC cells may involve an increase in the population of CSCs and that the aggressive tumorigenic properties of RT-R-TNBC cells are due to this increased population of CSCs. Therefore, in this study, we isolated CD24^−/low^/CD44^+^ cells from RT-R-MDA-MB-231 cells and determined the role of these CSCs in RT-R-TNBC on tumor progression and immune evasion.

Compared to MDA-MB-231 cells, RT-R-MDA-MB-231 cells exhibited increased proliferation, migration and invasion abilities, and these abilities were the most enhanced in CD24^−/low^/CD44^+^ cells isolated from RT-R-MDA-MB-231 cells (Figure 1). In addition, CD24^−/low^/CD44^+^ cells exhibited a reduced E-cadherin level but induction of N-cadherin, β-catenin, and Snail expression, suggesting that the EMT process is upregulated in CD24^−/low^/CD44^+^ cells. Moreover, HIF-1α/LOX induction and ESM-1 expression, which are involved in tumor metastasis and progression, were enhanced in CD24^−/low^/CD44^+^ cells compared to TNBC and RT-R-TNBC cells (Figure 2). Indeed, in the in vivo xenograft model, CD24^−/low^/CD44^+^ cell-injected mice exhibited greatly increased tumor growth and lung metastasis, accompanied by upregulated expression of EMT-related proteins and increased HIF-1α and ESM-1 levels in tumors and circulating LOX levels in plasma (Figure 5 and Figure 6). Therefore, we confirmed that the increased population of CSCs in breast cancer through the acquisition of RT resistance contributes to tumor development and metastasis.

The TME is a unique environment composed of proliferating tumor cells, the tumor stroma, blood vessels, infiltrating immune cells, and various tissue-associated cells, and it regulates molecular and cellular events occurring in surrounding tissues through interactions with the host [45]. Interestingly, the TME exhibits an effective barrier to immune cell functions, since tumors are not passive targets for host immunity but actively downregulate all phases of the tumor immune response via various strategies and mechanisms, a process called tumor immune escape. The mechanisms underlying tumor escape from the host immune system involve several strategies, as follows. First, tumor cells interfere with the induction of antitumor immune responses by downregulating the expression of costimulatory molecules but upregulating the expression of death receptors/ligands on tumor cells or APCs and by altering T cell receptor signaling in T cells and inducing dendritic cell apoptosis in the TME. In addition, the effector cell function is suppressed by Tregs and MDSCs and by the induction of apoptosis in effector T cells in the TME. In addition, tumor cells downregulate the recognition signals for immune cells by mediating HLA molecule expression on the tumor and by suppressing NK cell activity in the TME [45,46]. In particular, we were interested in alterations in the functional activity of NK cells, since BCSCs are reported to be resistant to NK cell cytotoxicity and contribute to lung metastasis by regulating the expression of ligands for NK cell receptors [30]. NK cells play important roles in innate defenses against infection and in the control of tumor growth and metastasis. The modulation and induction of NK cell function is mediated by activating or inhibitory receptors on their surface, and recruitment of NK cells toward tissues and lymphoid organs is regulated by chemokines and chemokine receptors, directing NK cell subsets to specific sites [47]. Therefore, we investigated whether the increased population of CSCs in RT-R-TNBC regulates the NK cell-mediated immune response using in vitro and in vivo models. Recruitment of NK cells around MDA-MB-231 cells and CD24^−/low^/CD44^+^ cells was similar (Figure 3A). However, NK cells showed stronger downregulation of cytotoxic activity toward CD24^−/low^/CD44^+^ cells than toward MDA-MB-231 cells (Figure 3B). NK cells which recognized abnormal target cells directly secrete lytic granules containing perforin and granzymes at the immunological synapse, and perforin generates pores in target cell membranes, allowing granzymes to access the cytoplasm and induce the apoptosis of the target cell [31]. Indeed, we found an enhanced suppressive effect of CD24^−/low^/CD44^+^ cells on the cytotoxic activity of NK cells through investigation of the NK cell lytic granule components perforin and granzyme B: CD24^−/low^/CD44^+^ cells suppressed the secretion of perforin and granzyme B (Figure 3C,D). Since tumor cells can regulate recognition signals and functionally modulate the expression of ligands for receptors on NK cells to mediate NK cell activity [45], we further investigated the dissimilarity in the expression of HLA-E and MICA/B in TNBC, RT-R-TNBC, and CD24^−/low^/CD44^+^ cells. Interestingly, CD24^−/low^/CD44^+^ cells exhibited induced HLA-E expression but reduced MICA/B expression in both cultured tumor cells (Figure 4A,B) and in tumor tissue from the in vivo xenograft model (Figure 7B,C), suggesting that the cytotoxic capacity of tumor-infiltrating NK cells may be strongly reduced by CSCs-rich RT-R-TNBC populations via alterations in signaling mediators for NK cells. While HLA-E can bind to both the inhibitory receptor NKG2A and the activating receptors NKG2C and NKG2E on NK cells, acting as a stimulant to suppress or enhance, respectively, NK cell cytotoxicity, the affinity of HLA-E for NKG2A and NKG2E is 6-fold higher than that for NKG2C [48]. In addition, NK cells tend to express higher levels of NKG2A than NKG2C or NKG2E [49], suggesting that HLA-E may act mainly as a suppressor of NK cell activation. In addition, the sheddase ADAM-10 mediates the reduction of NKG2D ligand expression on the cancer cell surface by cleaving MICA/B and is involved in the immune escape mechanism, which impairs NK cell recognition [33]. In this study, we observed increased levels of the active form of ADAM-10 in RT-R-MDA-MB-231 and CD24^−/low^/CD44^+^ cells compared to MDA-MB-231 cells (Figure 4C) and in tumor tissue from CD24^−/low^/CD44^+^ cell-injected mice (Figure 7D), suggesting that the augmented production of active ADAM-10 may be associated with the reduction in MICA/B levels in CSCs-rich RT-R-TNBC and the functional suppression of NK cells. Recently, ADAM-10 has been studied in breast cancer, especially in the human epidermal growth factor receptor 2 (HER2)-enriched subtype, and is considered a major sheddase of the HER2 receptor ectodomain [50], which contributes to HER receptor activation and induction of anticancer drug resistance [51]. Interestingly, ADAM-10 is reported to be involved in the oncogenic process and chemoresistance of TNBC by regulating Notch-1 and CD44 [52]. As mentioned above, we previously elucidated that RT-R-TNBC displaying an enhanced capacity for tumor progression exhibited increased Notch and CD44 levels [15]. Therefore, we expect that ADAM-10 also plays an important role in the acquisition of RT resistance in TNBC and plan to further study the mechanism by which the level of the ADAM-10 active form is increased in CSCs-rich RT-R-TNBC. Furthermore, MDSCs play a key role in immune suppression in cancer by regulating the activation and function of effector immune cells, and they also play roles in tumor angiogenesis, drug resistance, and metastasis [53]. MDSCs contribute to the enhancement of an immunosuppressive environment [54], and accumulated MDSCs can negatively regulate NK cell activation and function [55]. In this study, we also observed increased MDSC accumulation in the RT-R-MDA-MB-231 cell-injected group compared to the MDA-MB-231 cell-injected group and further enhancement of infiltration in the CD24^−/low^/CD44^+^ cell-injected group (Figure 7E). Therefore, we further plan to investigate how CSCs-rich RT-R-TNBC cells induce the recruitment and regulate the functions of various immune cells compared to TNBC cells.

In conclusion, we determined that tumors treated with anticancer therapy, especially RT-R breast tumors, have a larger population of BCSCs that exhibit more aggressive properties in terms of tumor growth and metastasis. Furthermore, BCSCs decrease the cytotoxicity of NK cells against themselves through alterations in signaling mediators for NK cell activation and thus can increase their survival, which may lead to tumor recurrence and diminish the efficacy of anticancer therapy. Therefore, we suggest that a therapeutic approach targeting CSCs in anticancer therapy-resistant breast cancer cell populations could solve the problem of immune evasion in RT-R tumors and ultimately improve the treatment of breast cancer patients.

## 4. Materials and Methods

### 4.1. Cell Culture and Establishment of RT-R-BC Cells

The human breast cancer cell line MDA-MB-231 was purchased from the Korea Cell Line Bank (Seoul, Korea), and RT-R-MDA-MB-231 cells were established as previously described [15]. In brief, MDA-MB-231 cells were irradiated with X-rays (2 Gy/25 fractions) to a total dose of 50 Gy, a commonly used clinical RT regimen in breast cancer patients. RT-R-MDA-MB-231 cells were used through 5 passages. The human umbilical endothelial cell line EA.hy926 and human NK cell line NK-92MI were obtained from the American Type Tissue Culture Collection (ATCC; Manassas, VA, USA). Breast cancer cell lines and endothelial cell lines were grown in RPMI-1640 medium and DMEM, respectively, supplemented with 10% FBS (GenDEPOT, Katy, TX, USA) and 1% penicillin and streptomycin (HyClone; Danaher Corporation, WA, Washington, DC, USA) at 37 °C in humidified air containing 5% CO_2_. NK-92MI cells were grown in alpha-MEM containing 2 mM L-glutamine, 1.5 g/L sodium bicarbonate (#12561-056, GIBCO; Thermo Fisher Scientific, Waltham, MA, USA) supplemented with 0.2 mM inositol (#I7508, Sigma-Aldrich, St. Louis, MO, USA), 0.1 mM 2-mercaptoethanol (#M3148, Sigma-Aldrich), 0.02 mM folic acid (#F8758, Sigma-Aldrich), 12.5% horse serum (GIBCO), and 12.5% FBS without ribonucleosides and deoxyribonucleosides at 37 °C in humidified air containing 5% CO_2_.

### 4.2. Isolation of CD24^−/low^ CD44^+^ Cells from Breast Cancer Cells

As BCSCs, CD24^−/low^ CD44^+^ cells were isolated from RT-R-MDA-MB-231 cells with a CD24 MicroBeads Kit (#130-095-951) and a CD44 MicroBeads Kit (#130-095-194) obtained from Miltenyi Biotec (Bergisch Gladbach, Germany). In brief, 1.5 × 10^7^ RT-R-MDA-MB-231 cells were magnetically labeled with CD24-Biotin and Anti-Biotin MicroBeads and were then separated on a MACS column in a MACS Separator (Miltenyi Biotec). Then, negatively isolated CD24^−/low^ RT-R-MDA-MB-231 cells were again magnetically labeled with CD44-Biotin MicroBeads and separated on a MACS column in a MACS Separator according to the manufacturer’s instructions.

### 4.3. Flow Cytometric Analysis to Compare the BCSC Population between MDA-MB-231 Cells and RT-R-MDA-MB-231 Cells

MDA-MB-231 and RT-R-MDA-MB-231 cells (1.4 × 10^6^) and isolated CD24^−/low^/CD44^+^ cells (1.4 × 10^6^) were stained with an APC-conjugated human anti-CD24 antibody (#FAB5247A, R&D Systems, Minneapolis, MN, USA) and a PE-conjugated human anti-CD44 antibody (#FAB3660P, R&D systems). Each conjugated antibody (7 μL) was added to 100 μL of the cell solution and was then incubated in the dark at 4 °C for 15 min. After incubation, the labeled cells were washed twice, suspended in 500 μL of 1 × PBS, and analyzed by flow cytometry using a FACSCanto II system (BD Biosciences, Franklin Lakes, NJ, USA). The population of CD24^−/low^/CD44^+^ cells in each cell group was quantified by analysis for 5 s at the same flow rate.

### 4.4. Cell Proliferation Assay

A total of 5000 cells/100 µL were seeded in 96-well plates, and 10 µL of the D-Plus™ CCK cell viability assay kit reagent (#CCK-3000, Dongin Biotech, Seoul, Korea) was added to each well at 0, 24, 48, and 72 h after seeding. After incubation for 30 min at 37 °C in the dark, the optical density of each well was measured using a microplate reader at a wavelength of 450 nm.

### 4.5. Colony Formation Assay

Five hundred cells were seeded in 6-well plates and incubated at 37 °C in humidified air containing 5% CO_2_. The culture medium was substituted with a fresh complete medium every 2–3 days for approximately 2 weeks. The colonies were fixed with methanol at room temperature for 10 min and completely dried after the methanol was discarded. Then, the colonies were stained with 0.1% Giemsa staining solution (#32884, Sigma-Aldrich, St. Louis, MO, USA) diluted with distilled water for 10 min at room temperature and counted.

### 4.6. Cancer Cell Migration and Matrigel-Invasion Assays

A total of 2 × 10^5^ cells/500 µL were seeded in 24-well plate inserts (#353097, 8 µm pore size membrane, Corning, Corning, NY, USA) for the migration assay or in endothelial cell-Matrigel Matrix (#356234, Corning)-coated 24-well plate inserts for the invasion assay. Then, 500 µL of complete medium was added to the lower chambers of the 24-well plates. After incubation for 20 h at 37 °C, the uninvaded cells remaining in the inserts were removed by scrubbing. The cells that invaded across the insert membranes were fixed with 4% formaldehyde in distilled water and permeabilized with 0.1% Triton X-100 in 1× PBS for 5 min at 4 °C. Then, the invaded cells were stained with 2 µg/mL 4′,6-diamidino-2-phenylindole dihydrochloride (DAPI, #D8417, Sigma-Aldrich) solution in distilled water for 30 min at room temperature in the dark and counted under a fluorescence microscope.

### 4.7. Protein Extraction and Western Blot Analysis

Cells were harvested, washed with ice-cold 1× PBS, and lysed with a radioimmunoprecipitation assay (RIPA) buffer [0.1% nonyl phenoxylpolyethoxylethanol-40 (NP-40) and 0.1% sodium dodecyl sulfate (SDS) in phosphate-buffered saline (PBS)] containing a protease inhibitor cocktail for 1 h on ice. Then, the suspension was centrifuged at 13,000 rpm for 15 min at 4 °C, and the supernatant (protein extract) was obtained. Approximately 30–80 µg of protein was subjected to 8% SDS-polyacrylamide gel electrophoresis (PAGE) and transferred onto polyvinylidene fluoride (PVDF) membranes. After blocking with 5% nonfat milk in TBS-T, membranes were incubated with anti-E-cadherin (#sc-7870, 1:1000, Santa Cruz Biotechnology, Dallas, TX, USA), anti-N-cadherin (#76011, 1:1000, Abcam, Cambridge, UK), anti-β-catenin (#sc-7199, 1:2000, Santa Cruz Biotechnology), anti-Snail (#3895, 1:1000, Cell Signaling, Danvers, MA, USA), anti-ESM-1 (#ab103590, 1:1000, Abcam), anti-HIF-1α (#ab2185, 1:1000, Abcam), anti-MICA/B (#ab203679, 1:1000, Abcam), anti-HLA-E (#ab2216, 1:1000, Abcam), anti-ADAM-10 (#ab1997, 1:1000, Abcam), and anti-β-actin (#MA5-15739, Thermo Fisher Scientific) antibodies for 1–2 days at 4 °C. Bound antibodies were detected with horseradish peroxidase (HRP)-conjugated secondary antibodies and an enhanced chemiluminescence (ECL) Western blotting detection reagent (#170-5061, Bio-Rad Laboratories, Hercules, CA, USA). For the detection of released LOX proteins, cells were incubated in serum-free medium for 16 h, and equal volumes of conditioned medium from each cell culture were then concentrated 40-fold through a centrifugal filter (#UFC801096, Millipore, Burlington, MA, USA) at a fixed angle (35°) by centrifugation at 7500× *g* for 25 min at 4 °C. The concentrated proteins were subjected to 8% SDS-PAGE and transferred onto PVDF membranes. After blocking with 5% nonfat milk in TBS-T, membranes were incubated with an anti-LOX (#sc-32409; 1:1000, Santa Cruz Biotechnology) antibody overnight at 4 °C, and bound antibodies were detected with HRP-conjugated secondary antibodies and an ECL reagent. Ponceau S staining was performed at room temperature for 5 min using Ponceau S Solution (#P7170, Sigma-Aldrich) and used as a loading control.

### 4.8. Total RNA Extraction and Reverse Transcription-Polymerase Chain Reaction (RT-PCR)

Total RNA from cells was then extracted using TRIzol reagent (#15596018, Thermo Fisher Scientific) as described in the manufacturer’s protocol. All primers were obtained from Bioneer (Deajeon, Korea), and RT-PCR was performed using TOPscript One-step RT-PCR Drymix (#RT421, Enzynomics, Deajeon, Korea) according to the manufacturer’s instructions. The primer sets are listed in Table 1. Amplification was performed under the following conditions: 30 cycles of denaturation at 95 °C for 30 s, annealing at 58 °C for 30 s, and extension at 72 °C for 1 min.

### 4.9. NK Cell Migration Assay

MDA-MB-231 cells and CD24^−/low^/CD44^+^ cells isolated from RT-R-MDA-MB-231 cells were cultured for 3 days. Then, the conditioned medium was obtained, and 500 μL of the conditioned medium from each cell culture was added to NK-92MI cells (2 × 10^5^ cells) seeded in 24-well plate inserts (#353097, 8 µm pore size membrane, Corning). After incubation for 24 h at 37 °C, NK cells that migrated into the lower chamber were counted by a trypan blue exclusion assay.

### 4.10. Cytotoxicity Assay

The cytotoxicity assay was performed using a CytoTox 96^®^ Non-Radioactive Cytotoxicity Assay Kit (#G1780, Promega, Madison, WI, USA). In brief, MDA-MB-231 cells and CD24^−/low^/CD44^+^ cells isolated from RT-R-MDA-MB-231 cells (5000 cells/60 μL were seeded in round-bottom 96-well plates, and NK-92MI cells were then added at the indicated ratios to the 96-well plates containing cancer cells. After coincubation for 4 h at 37 °C, the supernatants were obtained, and the level of lactate dehydrogenase (LDH) released into the supernatants was measured as described in the manufacturer’s protocol. Cytoxicity was quantified as follows:(1)$$\mathrm{Percent}\;\mathrm{cytotoxicity}=\frac{\mathrm{Experimental}\;\mathrm{LDH}\;\mathrm{Release}\;\left(\mathrm{OD}490\right)}{\mathrm{Maximum}\;\mathrm{LDH}\;\mathrm{Release}\;\left(\mathrm{OD}490\right)}\times 100$$

### 4.11. Measurement of Extracellular Perforin and Granzyme B Levels

The extracellular levels of perforin and granzyme B were measured with a Human Perforin ELISA Kit (#MBS9135839, MyBioSource, San Diego, CA, USA) and Human Granzyme B ELISA Kit (#MBS1777361, MyBioSource), respectively, according to the manufacturer’s instructions. MDA-MB-231 cells and CD24^−/low^/CD44^+^ cells isolated from RT-R-MDA-MB-231 cells were seeded in round-bottom 96-well plates (5000 cells/60 µL), and NK-92MI cells (target cell:NK cell ratio = 1:20) were added to the 96-well plates containing the cancer cells. After coincubation for 4 h at 37 °C, cell supernatants were obtained by removing the cells, and the perforin and granzyme B levels in the cell supernatants were then measured.

### 4.12. In Vivo Animal Study

Female BALB/c nude mice (6 weeks old, weighing 17–18 g) were obtained from OrientBio (Gyeonggi-do, Korea). Animals were maintained under the following environmental conditions: 22–26 °C; 40–60% humidity; 12 h light/dark cycle; and free access to sterilized food and water. Mice were subjected to whole-body gamma irradiation (5 Gy) prior to inoculation of human cancer cells, and after overnight, breast cancer cells (5 × 10^6^ cells/100 µL) were injected subcutaneously (n = 8 mice/group). Body weights and tumor volumes were measured three times weekly, beginning on the 7th day after injection [tumor volume (mm^3^) = 4/3 × π × width/2 × depth/2 × height/2]. At the end of the 40th day, mice were sacrificed, and the plasma LOX levels in blood samples were determined by Western blot analysis. The incidence of lung metastasis was determined by counting the metastatic foci on the lung surfaces. The tumor tissues were fixed with 10% formalin solution prior to paraffin infiltration and embedding. Sections (5 µm thick) were mounted onto MAS-GP type A Coated Slides (#S9911, Matsunami, Osaka, Japan), and immunohistochemical (IHC) analysis was then performed using anti-E-cadherin (#sc-7870, 1:100, Santa Cruz Biotechnology), anti-N-cadherin (#sc-7939, 1:50, Santa Cruz Biotechnology), anti-β-catenin (#sc-7199, 1:50, Santa Cruz Biotechnology), anti-Snail (#ab180714, 1:250, Abcam), anti-ESM-1 (#ab103590, 1:50, Abcam), anti-HIF-1α (#ab2185, 1:100, Abcam), anti-CD49b (#ab133557, 1:100, Abcam), anti-GR-1 (#RB6-8C5, 1:50, BioLegend, San Diego, CA, USA), anti-MICA/B (#ab203679, 1:100, Abcam), anti-HLA-E (#ab2216, 1:100, Abcam), and anti-ADAM-10 (#ab1997, 1:50, Abcam) antibodies. HRP-conjugated secondary antibodies were used, and IHC staining was then performed using a VECTASTAIN^®^ Elite^®^ ABC HRP Kit (#PK-6100, Vector Labs, Burlingame, CA, USA) and 3,3′-diaminobenzidine (DAB, #112080250, Thermo Fisher Scientific) according to the manufacturer’s instructions. Then, the sections were counterstained with Mayer’s hematoxylin (#CM3953, Cancer Diagnostics, Durham, NC, USA) and observed under a light microscope. The animal experimental protocol was approved by the Institutional Animal Care and Use Committee at Gyeongsang National University (approval number: GNU-200603-M0030), and all of the experiments were performed in compliance with the established institutional guidelines.

### 4.13. Statistical Analysis

All data were statistically analyzed using the GraphPad Prism 7 software (GraphPad Software, San Diego, CA). One-way ANOVA followed by Tukey’s post hoc test was carried out to compare differences among groups. The data are presented as the mean ± standard deviation (SD) values.

## Figures and Tables

**Figure 1 ijms-22-09639-f001:**
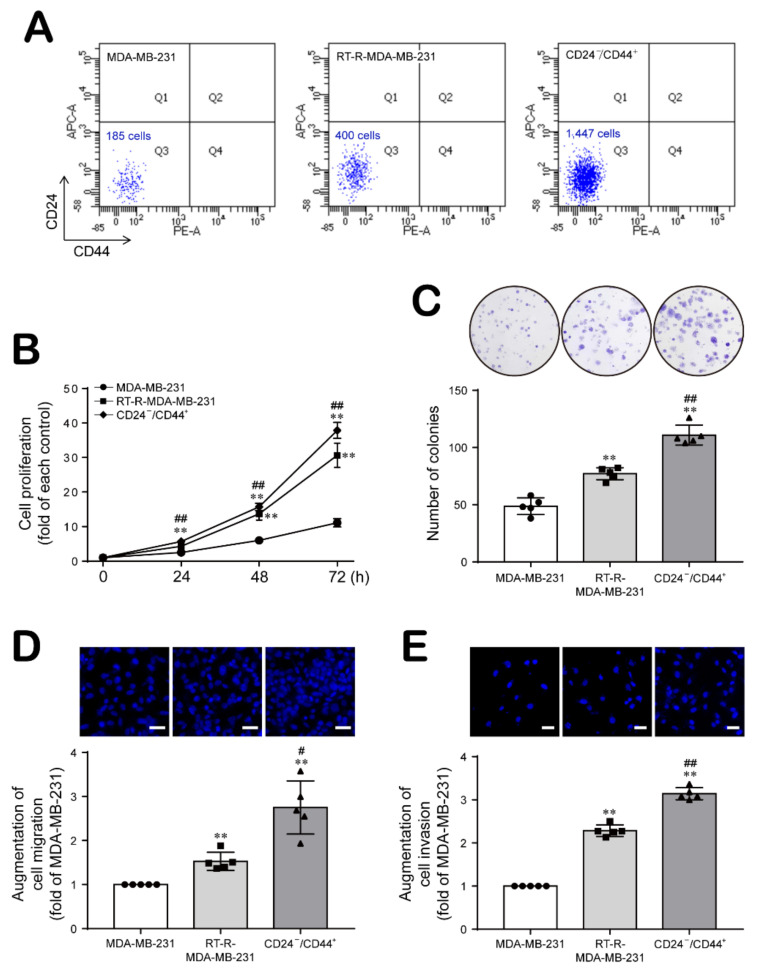
The larger population of CD24^−/low^/CD44^+^ BCSCs in RT-R-breast cancer cells shows increased proliferation, colony formation, migration and invasion abilities. (**A**) CD24^−/low^/CD44^+^ BCSCs were isolated from RT-R-MDA-MB-231 cells as described in the Materials and Methods section, and the CD24^−/low^/CD44^+^ BCSC populations among MDA-MB-231 and RT-R-MDA-MB-231 cells were compared to the isolated CD24^−/low^/CD44^+^ BCSCs. The same number (1 × 10^5^ cells) of MDA-MB-231, RT-R-MDA-MB-231, and CD24^−/low^/CD44^+^ cells isolated from RT-R-MDA-MB-231 cells were labeled with both an APC-conjugated anti-CD24 antibody and a PE-conjugated anti-CD44 antibody, and the population of CD24^−/low^/CD44^+^ cells was then quantified by flow cytometry for 5 s at the same flow rate as described in the Materials and Methods section. (**B**) Cell proliferation was measured 0, 24, 48, and 72 h after seeding using a CCK-8 assay kit as described in the Materials and Methods section. (**C**) Cells seeded in 6-well plates were cultured for approximately 2 weeks with the medium replaced with fresh complete medium every 2–3 days. Then, the colony formation ability was determined as described in the Materials and Methods section. (**D**,**E**) Cells were seeded on an 8 μm-pore size insert with a Matrigel-coated membrane for the migration assay (**D**) or with an endothelial cell-Matrigel-coated membrane for the invasion assay (**E**). After 20 h of incubation, the migrated (**D**) or invaded cells (**E**) were stained with DAPI and counted under a fluorescence microscope. The values are presented as the mean ± SD of five independent experiments. ** *p* < 0.01 compared to MDA-MB-231 cells; ^#^
*p* < 0.05, ^##^
*p* < 0.01 compared to RT-R-MDA-MB-231 cells.

**Figure 2 ijms-22-09639-f002:**
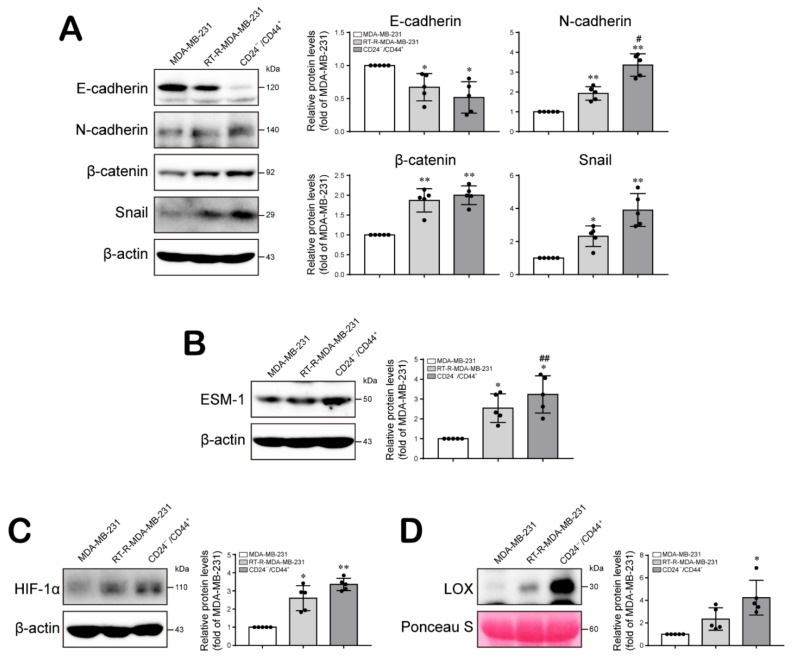
CD24^−/low^/CD44^+^ BCSCs increase the expression levels of tumor progression-related proteins and decrease the expression of NK cell function-related ligands on cancer cells. (**A**–**C**) The protein expression levels of EMT-related molecules (E-cadherin, N-cadherin, β-catenin, and Snail) (**A**), a tumor progression-related molecule (ESM-1) (**B**) and metastasis-related molecules (HIF-1α and LOX) (**C**,**D**) were determined in whole-cell lysates or conditioned medium of MDA-MB-231, RT-R-MDA-MB-231, and CD24^−/low^/CD44^+^ cells by Western blotting as described in the Materials and Methods section. β-Actin detection and Ponceau S staining were used as loading controls. The values are presented as the mean ± SD of five independent experiments. * *p* < 0.05, ** *p* < 0.01 compared to MDA-MB-231 cells; ^#^
*p* < 0.05, ^##^
*p* < 0.01 compared to RT-R-MDA-MB-231 cells.

**Figure 3 ijms-22-09639-f003:**
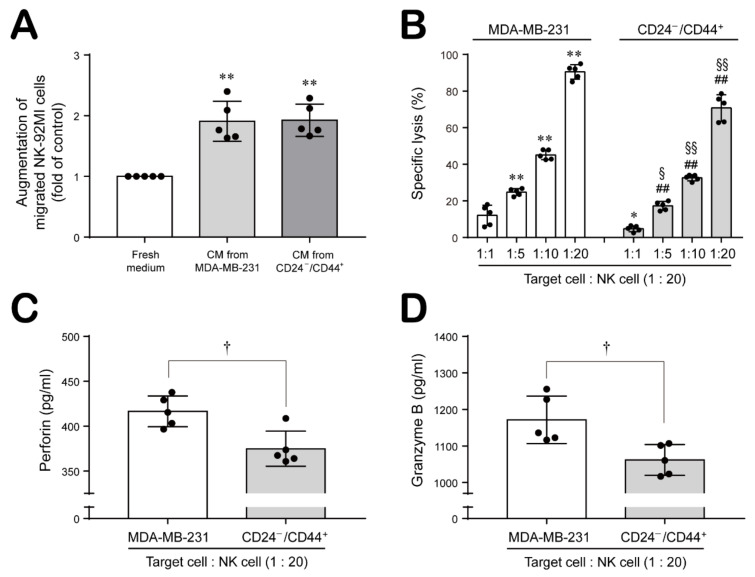
NK cells show lower cytotoxicity against CD24^−/low^/CD44^+^ BCSCs than against MDA-MB-231 cells. (**A**) NK-92MI cells were seeded into an 8 μm-pore size insert, and the conditioned medium (CM) obtained from MDA-MB-231 cells and CD24^−/low^/CD44^+^ cells isolated from RT-R-MDA-MB-231 cells was added to the lower chamber. After incubation for 24 h, the migrated NK cells were counted by a trypan blue exclusion assay as described in the Materials and Methods section. The values are presented as the mean ± SD of five independent experiments. ^**^
*p* < 0.01 compared to the fresh medium. (**B**–**D**) MDA-MB-231 cells and CD24^−/low^/CD44^+^ cells isolated from RT-R-MDA-MB-231 cells were seeded in round-bottom 96-well plates, and NK-92MI cells were then added at the indicated ratios (1:1, 1:5, 1:10, 1:20) to 96-well plates containing cancer cells. After coincubation for 4 h, LDH (**B**), perforin (**C**), and granzyme B (**D**) were measured in cell supernatants as described in the Materials and Methods section. The values are presented as the mean ± SD of five independent experiments. * *p* < 0.05, ** *p* < 0.01 compared to the control group (1:1) of MDA-MB-231 cells; ^##^
*p* < 0.01 compared to the control group (1:1) of CD24^−/low^/CD44^+^ cells; ^§^
*p* < 0.05, ^§§^
*p* < 0.01 compared to each same ratio point of NK-92MI cells to MDA-MB-231 cells; ^†^
*p* < 0.05 compared between MDA-MB-231 cells and CD24^−/low^/CD44^+^ cells.

**Figure 4 ijms-22-09639-f004:**
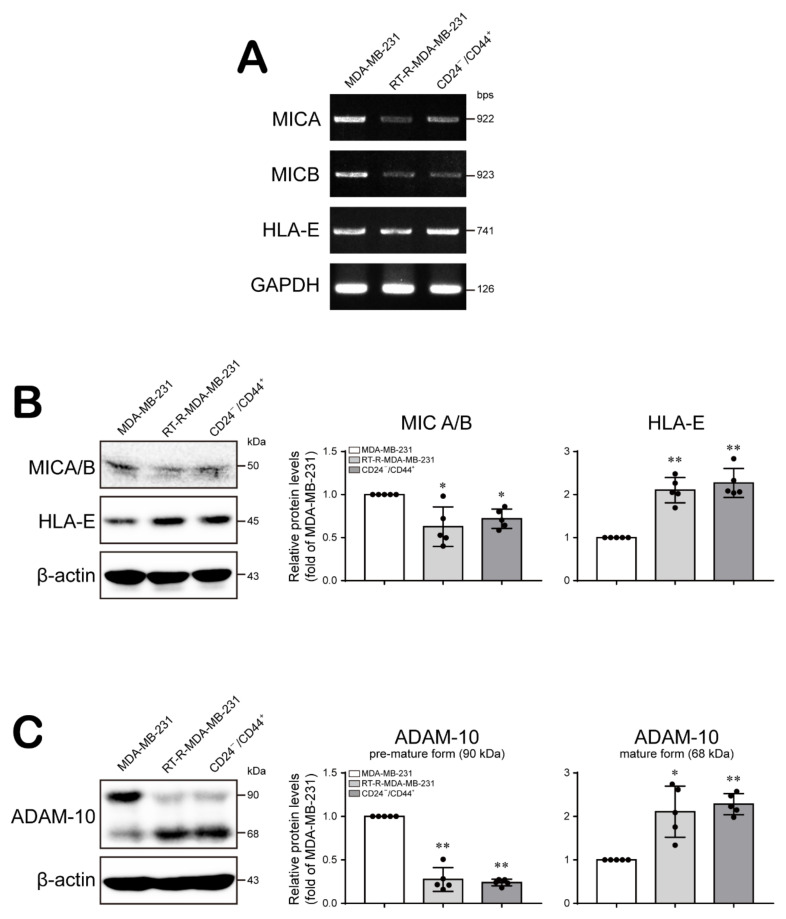
CD24^−/low^/CD44^+^ BCSCs modulate the expression of sheddase, ADAM-10, and cell surface ligands for receptors of NK cells to regulate NK cell cytotoxicity. (**A**,**B**) The mRNA and protein expression levels of MICA/B and HLA-E were determined in MDA-MB-231, RT-R-MDA-MB-231, and CD24^−/low^/CD44^+^ cells by RT-PCR (**A**) and Western blotting (**B**) as described in the Materials and Methods section. (**C**) The protein levels of the premature and mature forms of ADAM-10 were examined in MDA-MB-231, RT-R-MDA-MB-231, and CD24^−/low^/CD44^+^ cells by Western blotting. GAPDH and β-actin were detected as loading controls. The values are presented as the mean ± SD of five independent experiments. * *p* < 0.05, ** *p* < 0.01 compared to MDA-MB-231 cells.

**Figure 5 ijms-22-09639-f005:**
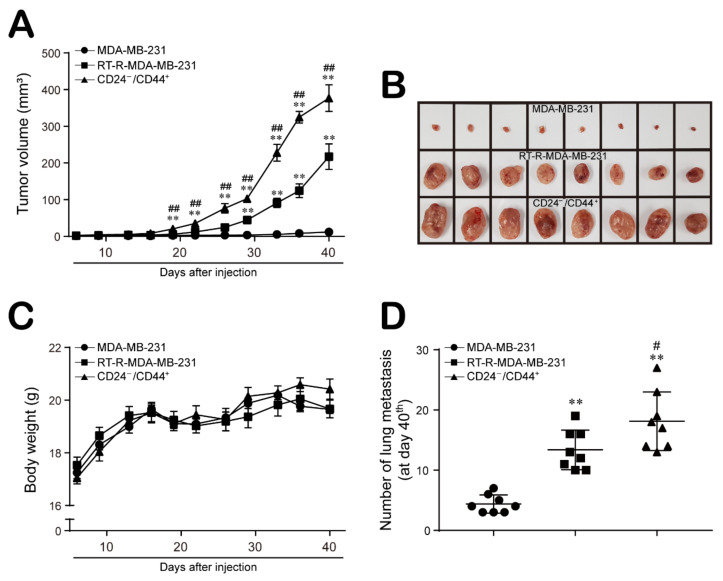
CD24^−/low^/CD44^+^ BCSCs show enhanced aggressive properties in terms of tumor growth and metastasis compared to MDA-MB-231 and RT-R-MDA-MB-231 in an in vivo mouse model. Six-week-old female BALB/c nude mice were injected subcutaneously with 5 × 10^6^ MDA-MB-231, RT-R-MDA-MB-231 or CD24^−/low^/CD44^+^ cells (n = 8 mice/group). Beginning on the 7th day after injection, tumor volumes (**A**,**B**) and body weights (**C**) were measured three times weekly during the tumor development. On the 40th day, the mice were sacrificed, and the incidence of lung metastasis (**D**) was determined. ** *p* < 0.01 compared to the MDA-MB-231 cell-injected group; ^#^
*p* < 0.05, ^##^
*p* < 0.01 compared to the RT-R-MDA-MB-231 cell-injected group.

**Figure 6 ijms-22-09639-f006:**
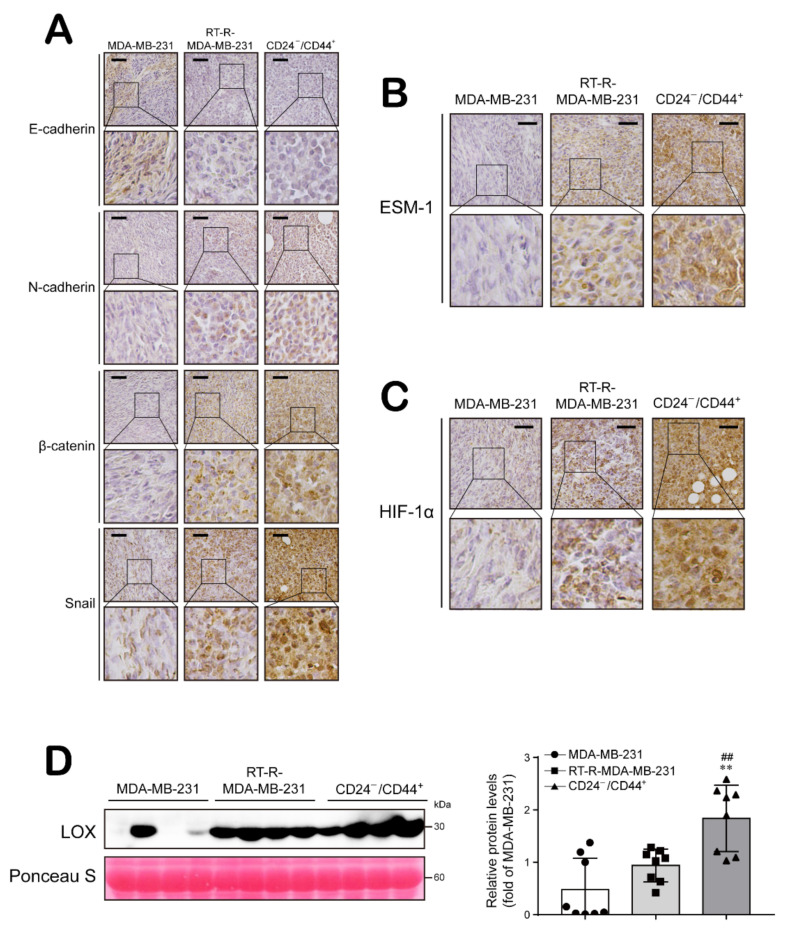
The expression of tumor progression- and metastasis-related molecules is highly upregulated in the tumor tissues of CD24^−/low^/CD44^+^ BCSC-injected mice. (**A**–**C**) Tumor tissue sections were immunohistochemically stained with antibodies specific for the indicated proteins: The EMT-related proteins E-cadherin, N-cadherin, β-catenin, and Snail (**A**); the tumor progression-related protein ESM-1 (**B**); and the metastasis-related protein HIF-1α (**C**). Then, the stained sections were counterstained with Mayer’s hematoxylin solution. Scale bar: 100 µm. (**D**) The plasma LOX levels in blood samples were determined by Western blot analysis (n = 8 mice/group). ** *p* < 0.01 compared to the MDA-MB-231 cell-injected group; ^##^
*p* < 0.01 compared to the RT-R-MDA-MB-231 cell-injected group.

**Figure 7 ijms-22-09639-f007:**
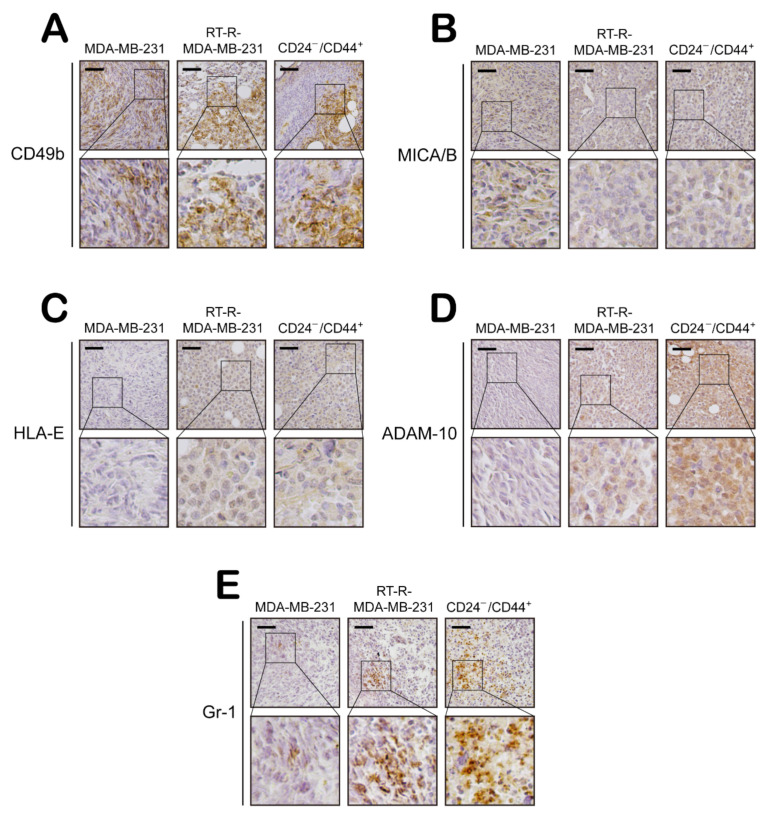
NK cells infiltrate tumor tissues but are functionally impaired through a reduction in the expression of the activating ligand MICA/B and induction of the inhibitory ligand HLA-E in CD24^−/low^/CD44^+^ BCSC-injected mice. Tumor tissue sections from MDA-MB-231 cell-, RT-R-MDA-MB-231 cell-, and CD24^−/low^/CD44^+^ BCSC-injected mice were immunohistochemically stained with antibodies specific for the indicated proteins: The murine NK cell marker CD49b (**A**), the NKG2D ligand MICA/B (**B**), the NKG2A ligand HLA-E (**C**), the MICA/B sheddase ADAM-10 (**D**), and the murine MDSC marker Gr-1 (**E**). Then, the stained sections were counterstained with Mayer’s hematoxylin solution. Scale bar: 100 µm.

**Table 1 ijms-22-09639-t001:** Sequences of primers used in the study.

Gene	Primer Sequence
hMICA	Forward: 5′-TCCTGCTTCTGGCTGGCATC-3′ Reverse: 5′-GCAGAAACATGGAATGTCTG-3′
hMICB	Forward: 5′-CTGCTGTTTCTGGCCGTCGC-3′ Reverse: 5′-GAAACATATGGAAAGTCTGTC-3′
hHLA-E	Forward: 5′-GTGAATCTGCGGACGCTGCG-3′ Reverse: 5′-CTTAGAGTAGCTCCCTCCTT-3′
hGAPDH	Forward: 5′-TCAACAGCGACACCCACTCC-3′ Reverse: 5′-TGAGGTCCACCACCCTGTTG-3′

## Data Availability

All data are available via the corresponding author.

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
