# Peer review of "NK Cells Lose Their Cytotoxicity Function against Cancer Stem Cell-Rich Radiotherapy-Resistant Breast Cancer Cell Populations"

_ijms, 2021, doi:10.3390/ijms22179639_

Round 1

Reviewer 1 Report

This is an important and very interesting paper on the mutual interactions between the population of tumor cells and immunological system during radiotherapy. I have two questions and several small technical improvements to suggest

  1. Because of the importance and controversy about CSC, the topic deserves more support. Please cite also e.g. Boiko et al., 2010   doi: 10.1038/nature09161
  2. Did you consider the effect of cell size on the observed changes in the tempo of tumor growth? This simple physical factor may also affect the progression (see Klos & Plonka 2020, DOI: https://doi.org/10.18388/abp.2020_5500
  3. line 96 - types of cells
  4. line 156, 167 - stronger (insteadf of "more strongly")
  5. line 276 - expression of (a space is lacking)

Author Response

This is an important and very interesting paper on the mutual interactions between the population of tumor cells and immunological system during radiotherapy. I have two questions and several small technical improvements to suggest

  1. Because of the importance and controversy about CSC, the topic deserves more support. Please cite also e.g. Boiko et al., 2010, doi: 10.1038/nature09161

→ Answer: Thank you for your comment. Since the first modern evidence for the role for stem cells in cancer was reported in 1994 (Lapidot et al., 1994), the studies on CSCs have suggested the important roles of CSCs in tumor progression in various types of cancers. As we described in the manuscript, CSCs show a high self-renewal capacity, the propensity to differentiate into actively proliferating tumor cells, the ability to repair DNA damage, and low levels of reactive oxygen species (ROS), and the small population of CSCs may determine the biological properties of cancers, including the therapeutic response. Based on these reports, we added more references (see below and Reference 2~5 in the manuscript) including the reference you recommended to support the importance and controversy about CSCs and modified some sentences (please see lines 28~36). Thank you for your advice to enrich the content of this manuscript.

<References>

Lapidot, T.; Sirard, C.; Vormoor, J.; et al. A cell initiating human acute myeloid leukaemia after transplantation into SCID mice. Nature. 1994;367(6464):645–648.

Reya, T.; Morrison, S.J.; Clarke, M.F.; Weissman, I.L. Stem cells, cancer, and cancer stem cells. Nature 2001, 414, 105-111; DOI: 10.1038/35102167.

Beck, B.; Blanpain, C. Unravelling cancer stem cell potential. Nat. Rev. Cancer 2013, 13, 727-738; DOI: 10.1038/nrc3597.

Ayob, A.Z.; Ramasamy, T.S. Cancer stem cells as key drivers of tumour progression. J. Biomed. Sci. 2018, 25, 20; DOI: 10.1186/s12929-018-0426-4.

Boiko, A.D.; Razorenova, O.V.; van de Rijn, M.; Swetter, S.M.; Johnson, D.L.; Ly, D.P.; Butler, P.D.; Yang, G.P.; Joshua, B.; Kaplan, M.J.; Longaker, M.T.; Weissman, I.L. Human melanoma-initiating cells express neural crest nerve growth factor receptor CD271. Nature 2010, 466, 133-137; DOI: 10.1038/nature09161.

  1. Did you consider the effect of cell size on the observed changes in the tempo of tumor growth? This simple physical factor may also affect the progression (see Klos & Plonka 2020, doi: https://doi.org/10.18388/abp.2020_5500)

→ Answer: Thank you for your valuable comment. Based on our observations, these three types of breast cancer cells (MDA-MB-231, RT-R-MDA-MB-231 and CD24-/low CD44+ cells) do not differ in cell size (please see attached figure). Therefore, we regard that the cells size had no effect on in vivo tumor growth in this study.

  1. line 96 - types of cells

→ Answer: According to your comment, we changed CD24-/low CD44+ cells into CD24-/low CD44+ BCSCs (breast cancer stem cells) for defining the types of cells (page 3, lines 99~101).

  1. line 156, 167 - stronger (insteadf of "more strongly")

→ Answer: We correct them with ‘stronger’ according to your comment (page 5, line 156 and 171).

  1. line 276 - expression of (a space is lacking)

→ Answer: We put a space between ‘expression’ and ‘of’ according to your comment (page 10, line 277, 278). Thank you very much for your kind indications.

Reviewer 2 Report

We well know that genetic alterations, clonal evolution, and the tumor microenvironment promote cancer progression, metastasis and therapy resistance. These events favor a great phenotypic heterogeneity and plasticity of cancer cells that contribute to tumor progression and resistant disease. Targeting resistant cancer cells is a major challenge in oncology; however, the underlying processes and mechanisms are not yet understood.

Recent evidence also shows that stem-like phenotypes in cancer cells, promoted by cancer stem cells, are the major contributors to cancer recurrence, chemo-resistance and metastasis. In addition, through the simultaneous tracing of single-cell clonal origin and assessment of their proliferative and transcriptional states, researchers have shown that these persistent cells arise from different rare cell lineages with distinct transcriptional and metabolic programs.

In an in vivo model, mice injected with CD24-/low/CD44 + cells showed increased tumor progression and lung metastases through increased tumor progression-related molecules and altered host immune responses. NK cells were recruited again in the peritumor area of the tumor but lost their cytotoxicity because of the altered expression of activating and inhibitory ligands on the tumors.

These findings suggest CSCs can cause immune cells to evade the tumor, resulting in tumor progression. Experiments on mice have confirmed that cancer cells somehow educate Nk cells, which end up promoting the spread of cancer. When Nk cells come into contact with tumors, they undergo dramatic changes in gene expression. Natural killer cells kill by releasing the contents of their granules, small vesicles containing perforin and granzyme.

It is not enough for the killer cell to be in the right place at the right time to eliminate the intended victim. It must be able to use its own weapon. However, cancer cells often escape attempts at annihilation. By studying glioblastoma multiforme, US researchers have discovered what prevents NK cells from killing cancer cells. Cell culture experiments have shown that NK cells recognized and killed glioblastoma cells, including cancer stem cells. Comparing the NK cells present in glioblastomas with those present in the blood of healthy subjects, however, the Texan researchers found differences, as if the tumor could render NK harmless. From their experiments, they understood that this depended on a molecule, the cytokine TGFβ1, produced following contact between the two cell types. Using animal models, the researchers also showed that by acting on the TGFβ1 expression, it restored the activity of NK cells. These authors speculate that glioblastoma could be cured by administering NK cells from a donor along with a drug that blocks TGFβ1.

The conclusion based on all this information is that these rare lineages of the gene expression profiles and their metabolic characteristics could delay or even prevent a disease recurrence. Therefore, understanding persistent cell dynamics is critical for the design of more effective chemotherapies for cancer treatment.

The authors of the manuscript moved based on this knowledge. They analyzed cell divisions in breast cancer cells, monitoring the results for cell lineages of the tumor cell line. They found in a radio-resistant breast cancer cells a high proliferating population of CD24-/low/CD44+ cells with increased migration and invasion abilities. These cells show higher expression levels of tumor progression-related proteins and recruit NK cells. They also tested the NK cytotoxicity suppression through the modulation of the levels of NK cell function-related ligands. Their conclusion was that “CD24-/low/CD44+ cells exhibit enhanced aggressive features supporting the progression and metastasis of breast cancer in vivo”.

The interest of their research lies because they have shown that in TNBC too there is a resistance mechanism glioblastoma-like, although there is still a lot to know to characterize it. This opens a window to check if other metastasizing tumors show similar mechanisms. The experimental design is well set and experiments well performed.

Author Response

We well know that genetic alterations, clonal evolution, and the tumor microenvironment promote cancer progression, metastasis and therapy resistance. These events favor a great phenotypic heterogeneity and plasticity of cancer cells that contribute to tumor progression and resistant disease. Targeting resistant cancer cells is a major challenge in oncology; however, the underlying processes and mechanisms are not yet understood.

Recent evidence also shows that stem-like phenotypes in cancer cells, promoted by cancer stem cells, are the major contributors to cancer recurrence, chemo-resistance and metastasis. In addition, through the simultaneous tracing of single-cell clonal origin and assessment of their proliferative and transcriptional states, researchers have shown that these persistent cells arise from different rare cell lineages with distinct transcriptional and metabolic programs.

In an in vivo model, mice injected with CD24-/low/CD44 + cells showed increased tumor progression and lung metastases through increased tumor progression-related molecules and altered host immune responses. NK cells were recruited again in the peritumor area of the tumor but lost their cytotoxicity because of the altered expression of activating and inhibitory ligands on the tumors.

These findings suggest CSCs can cause immune cells to evade the tumor, resulting in tumor progression. Experiments on mice have confirmed that cancer cells somehow educate Nk cells, which end up promoting the spread of cancer. When Nk cells come into contact with tumors, they undergo dramatic changes in gene expression. Natural killer cells kill by releasing the contents of their granules, small vesicles containing perforin and granzyme.

It is not enough for the killer cell to be in the right place at the right time to eliminate the intended victim. It must be able to use its own weapon. However, cancer cells often escape attempts at annihilation. By studying glioblastoma multiforme, US researchers have discovered what prevents NK cells from killing cancer cells. Cell culture experiments have shown that NK cells recognized and killed glioblastoma cells, including cancer stem cells. Comparing the NK cells present in glioblastomas with those present in the blood of healthy subjects, however, the Texan researchers found differences, as if the tumor could render NK harmless. From their experiments, they understood that this depended on a molecule, the cytokine TGFβ1, produced following contact between the two cell types. Using animal models, the researchers also showed that by acting on the TGFβ1 expression, it restored the activity of NK cells. These authors speculate that glioblastoma could be cured by administering NK cells from a donor along with a drug that blocks TGFβ1.

The conclusion based on all this information is that these rare lineages of the gene expression profiles and their metabolic characteristics could delay or even prevent a disease recurrence. Therefore, understanding persistent cell dynamics is critical for the design of more effective chemotherapies for cancer treatment.

The authors of the manuscript moved based on this knowledge. They analyzed cell divisions in breast cancer cells, monitoring the results for cell lineages of the tumor cell line. They found in a radio-resistant breast cancer cells a high proliferating population of CD24-/low/CD44+ cells with increased migration and invasion abilities. These cells show higher expression levels of tumor progression-related proteins and recruit NK cells. They also tested the NK cytotoxicity suppression through the modulation of the levels of NK cell function-related ligands. Their conclusion was that “CD24-/low/CD44+ cells exhibit enhanced aggressive features supporting the progression and metastasis of breast cancer in vivo”.

The interest of their research lies because they have shown that in TNBC too there is a resistance mechanism glioblastoma-like, although there is still a lot to know to characterize it. This opens a window to check if other metastasizing tumors show similar mechanisms. The experimental design is well set and experiments well performed.

→ Answer: Thank you for your interest in our study and for your positive and knowledgeable comments. Now, we are planning to conduct more in-depth studies about the roles of BCSCs in acquirement of a resistance against anticancer therapy and in regulation of immune responses. Your comments will be of great help in conducting related research. We will take your comments into account as we conduct our research. Thank you again.
